# Ultrastructure of the Sensilla on the Antennae and Mouthparts of Bean Weevils, *Megabruchidius dorsalis* (Coleoptera: Bruchinae)

**DOI:** 10.3390/insects12121112

**Published:** 2021-12-13

**Authors:** Siyu Chen, You Li, Fangling Xu, Maofa Yang, Xiurong Wang, Chengxu Wu

**Affiliations:** 1College of Forestry, Guizhou University, Guiyang 550025, China; csyxhq@163.com (S.C.); flxu@gzu.edu.cn (F.X.); xrwang@gzu.edu.cn (X.W.); 2Vector-Borne Virus Research Center, Fujian Province Key Laboratory of Plant Virology, Fujian Agriculture and Forestry University, Fuzhou 350002, China; yourreason@hotmail.com; 3School of Forestry, Fisheries and Geomatics Sciences, University of Florida, Gainesville, FL 32611, USA; 4College of Tobacco Science, Guizhou University, Guiyang 550025, China; gdgdly@126.com; 5Institute of Entomology, College of Agriculture, Guizhou University, Guiyang 550025, China

**Keywords:** Bruchinae, *Gleditsia*, storage pest, chemoreceptors, mechanoreceptors, scanning electron microscopy

## Abstract

**Simple Summary:**

In this paper, we used scanning electron microscopy (SEM) to describe the morphological types, number of sensilla, and their distributions on the antennae and mouthparts of both sexes of the bean weevil, *Megabruchidius dorsalis* (Coleoptera: Bruchinae). The results showed twelve subtypes on antennal sensilla and five types of sensilla on maxillary and labial palps. No sexual dimorphism in sensilla type was observed, but there were variations between male and female in the numbers and distribution along with the antennae. In addition, we discussed potential function related to structure, through comparisons with previous studies of bruchids and other insects. This information will support further studies of semiochemicals as effective biological controls of this pest.

**Abstract:**

*Megabruchidius dorsalis* (Fåhraeus, 1839) (Coleoptera: Bruchinae) is an important pest that damages the seeds of *Gleditsia* L. (Fabaceae: Caesalpinioideae). This beetle searches for host plants with its sensory system. To further explore the mechanisms of host location and to understand the ultrastructure of *M. dorsalis*, we examined the morphology and distribution of its sensilla on the antennae and mouthparts of male and female adults, using scanning electron microscopy (SEM). Both male and female antennae are serrated and can be divided into scape, pedicel, and flagellum. There were seven types and eight subtypes of antennal sensilla, including Bőhm bristles (BB), two subtypes of sensilla trichoid (ST1, ST2), two subtypes of sensilla chaetica (SC1, SC2), four subtypes of sensilla basiconic (SB1, SB2, SB3, SB4), sensilla cavity (SCa), sensilla auricillica (SA), and sensilla gemmiformium (SG). Five types of maxillary and labial palp sensilla in the mouthparts were observed: sensilla chaetica (SC), sensilla trichoidea (ST), sensilla styloconica (SSt), sensilla coeloconica (SCo), and sensilla digitiform (SD). No sexual dimorphism in sensilla type was observed, but there were variations between males and females in the numbers and distribution along the antennae. There were more SA in males than in females, while the number of ST sensilla in the maxillary palps were lower in males than in females. ST1 were most abundant in both sexes. We discussed potential function related to structure via comparisons with previous investigations of bruchids and other insects. Our results provide a theoretical basis for further studies on sensory physiological function, using semiochemicals as effective biological controls of *M. dorsalis*.

## 1. Introduction

*Gleditsia* Lam., the genus in the family Fabaceae (Caesalpinioideae) is well known for significant medicinal uses and biological activities, including antimicrobial, anti-oxidant, and antischistosomal activities [1]. *Megabruchidius dorsalis* (Fåhraeus, 1839) (Coleoptera: Bruchinae) is a seed pest. It is an oligophagous insect, and its known host plants are most of the species within *Gleditsia* [2]. It is distributed in India, Japan, Papua New Guinea, and China, and was recently introduced into Europe [3]. In the introduced areas, most of the seeds of *Gleditsia* L.s are infested by *M. dorsalis* on the tree [4]. The females oviposit on the surface of seeds, and the larvae burrow into the seeds until emergence, making it difficult for chemicals to enter the beans to kill the pest [5].

As the chemical ecology of insects has developed, some insect behaviors have been found to be associated with their microanatomical features. Those behaviors include feeding, selecting mates [6], oviposition [7], and inter- and intra-species communication. These phenomena are triggered by chemical and physical stimuli from the surrounding environment and recognized by an insect’s major sensory structures (sensilla) [8]. The cephalosome of an insect, and especially the antennae, maxillary palps, and labial palps, have various sensory structures with function that include mechanoreception, gustation, olfaction, hygroreception, and thermoreception [9,10,11]. 

The morphological characterization and distribution of sensilla on the antennae and mouthparts reveal associated sensory modalities of the sensilla, and provide valuable information about ther behavioral responses of sensilla and the chemical ecology of insects [12]. Hu et al. [13] found that the sensilla of Bőhm bristles (BB) of *Echinothrips americanus* Morgan (Thysanoptera: Thripidae), sensilla trichoid and sensilla chaetica, these were common sensilla, the external morphology of which conforms to the overall characteristics of mechanical sensilla [14]. Crook et al. [15] observed that cross-sections of sensilla chaetica have a non-perforated walls and a single sensory neuron, with the dendrite terminating in the form of a tubular body, which suggested that these sensilla are typical tactile mechanoreceptors. Fukuda et al. [16] found that sensilla trichoid could also sense chemical stimulation (pheromone receptors) by electrophysiological methods; sensilla trichoid were assumed to have a sex pheromone receptive function in *Mamestra suasa* (Lepidoptera: Noctuidae) [17]. Hu et al. [13] found that the anatomical characteristics of sensilla chaetica proved that it is also contacting the chemoreceptor. Faucheux et al. [18] proposed that sensilla chaetica have a gustatory function to recognize mates by receptor contact sex pheromones. Thus, sensilla trichoid and sensilla chaetica are dual-function receptors. Fukuda et al. [16] inferred that the most probable function of the sensilla cavity in relation to *Callosobruchus rhodesianus* (Coleoptera: Chrysomelidae: Bruchinae) are chemo-, thermo-, or hygro-reception. Sensilla basiconic often have numerous wall pores, which indicates an olfactory function [8]. Sensilla basiconica were found to respond to host plant odors, and each of the subtypes can be specialized for a particular set of host and non-host volatiles in *Adelphocoris lineolatus,* Goeze (Hemiptera: Miridae) [19] and *Chlorophorus caragana* Xie & Wang (Coleoptera: Cerambycidae) [20]. Faucheux et al. [21] stated that characteristic ultiparous double-walled sensilla basiconica respond to short-chain n-acids and amines (odors). Except for the sensilla that were reported in the mentioned species, the external morphology of antennae and mouthparts, including the types, abundance, and distribution of the sensilla, have been well-studied recently in *Callosobruchus* spp. and *Acanthoscelides obtectus* [13,16,22,23,24]. However, no descriptions of the ultrastructure of *M. dorsalis* antennal and mouthparts sensilla have yet been reported.

In this paper, we used scanning electron microscopy (SEM) to describe the morphological types and number of sensilla and their distributions on the antennae and mouthparts (the maxillary and labial palps) in both males and females of adult *M. dorsalis*. In addition, we discussed potential function related to structure via comparisons with previous studies of bruchids and other insects. This information will support further research on mating and host-finding mechanisms, electrophysiology, and behavioral ecology, as well as on semiochemicals as effective biological controls for this beetle pest.

## 2. Materials and Methods

### 2.1. Insects

Adults of *M. dorsalis* were collected in October 2020 from *Gleditsia sinensis* Lam. in the vicinity of Zhijin County, Guizhou Province, China (106°03″~106°04″ E, 26°32″~26°33″ N) and reared on seeds of *G. sinensis* (from the vicinity of Zhijin County) in plastic boxes (225 × 155 × 80 mm) in a rearing incubator (RXZ–380A–LED, Ningbo Instrument Factory) at Forest Conservation Laboratory in college of Forestry, Guizhou University. The incubator was maintained at 28 ± 1 °C, 65 ± 5% RH, with a 14 h dark–10 h light photoperiod. After adults emerged, the beetles were separated by sex for further testing according to the morphological characteristics described by Li et al. [25].

### 2.2. Sample Preparation for Scanning Electron Microscopy (SEM)

We anesthetized adult *M. dorsalis* (six males and six females) at −18 °C, and then removed the heads) and antennae of three males and three females using scalpels under a stereomicroscope (XT2, Beijing Tech Instrument Co. Ltd., Beijing, China), and washed them for 3 min in 70% ethanol (Sinaopharm Group Chemical Reagent Co. Ltd., Shanghai, China) using an ultrasonic cleaner [26]. The samples were immediately fixed in 2.5% glutaraldehyde (Servicebio) for 12 h at 4 °C, then rinsed with 0.01 M phosphate-buffered saline (PBS) three times, each time for 15 min. The samples were then transferred into 1% OsO4 (Ted Pella Inc.) in a 0.1 M phosphate buffer (PB) (pH 7.4) (Servicebio) for 1–2 h at room temperature. After that, we washed each sample in 0.1 M PB (pH 7.4) three times for 15 min, then dehydrated each of them in a graded alcohol series (30, 50, 70, 80, 90, 95, and twice at 100%) for 15 min, and then in isoamyl acetate (Sinaopharm Group Chemical Reagent Co. Ltd., Shanghai, China) for 15 min. Subsequently, dried samples were processed with a critical point dryer (K850, Quorum), and the specimens were attached to metallic stubs using carbon stickers, then sputter-coated (HITACHI MC1000, High Technologies, Tokyo, Japan) with gold for 30–120 s. The examination was carried out using an SU8100 scanning electron microscope (HITACHI High Technologies, Tokyo, Japan) at 3.0 kV.

### 2.3. Statistical Analyses

The morphology, number, and distribution of sensilla on the antennae, maxillary palps, and labial palps of male and female *M. dorsalis* were identified, measured, and counted. We placed the antennae on a horizontal plane similar to that shown in Figure 1a, and counted the total number of sensors on both sides (Figure 1a and Figure 2a, rotated 180°). The same approach was followed for mouthparts, as shown in Figure 3a. The morphological terminology and classification of the sensilla used here followed that of Zacharuk [8] and Schneider [11], which determined the type and subtypes of sensilla based on sensilla morphology, size, and distribution. In addition, the SEM pictures were compared with previous reports to determine the type and subtypes of sensilla. The number of each type of sensilla was determined manually (counted and discriminated by one person to reduce the personal equation), and images were processed with Adobe Photoshop CC (Adobe Systems Software Ireland Ltd., San Jose, CA, USA). The width and length of each antenna segment (there were 11 segments per antenna) and the length of the sensilla were measured. When the number of sensillum on an antenna was more than 10, the 10-length data for this type of sensilla were selected randomly, or otherwise measured along the length of all such sensilla (three adults per sex) using AutoCAD2014 software (Autodesk, San Rafael, CA, USA) (Figure 1). The production of data analyses was performed using SPSS 26.0 software (International Business Machines Corporation, IBM, Armonk, NY, USA). The differences in the length and number of sensilla were analyzed by a nonparametric test (the Kruskal-Wallis test), and comparisons between the sexes were made using independent-samples *t*-test. Before analyses, the data were normally distributed by the Kolmogorov-smirnov test. Data were reported as having means of ± S.E. or medians (min–max). The level of significance in all tests was set at 0.05.

**Figure 1 insects-12-01112-f001:**
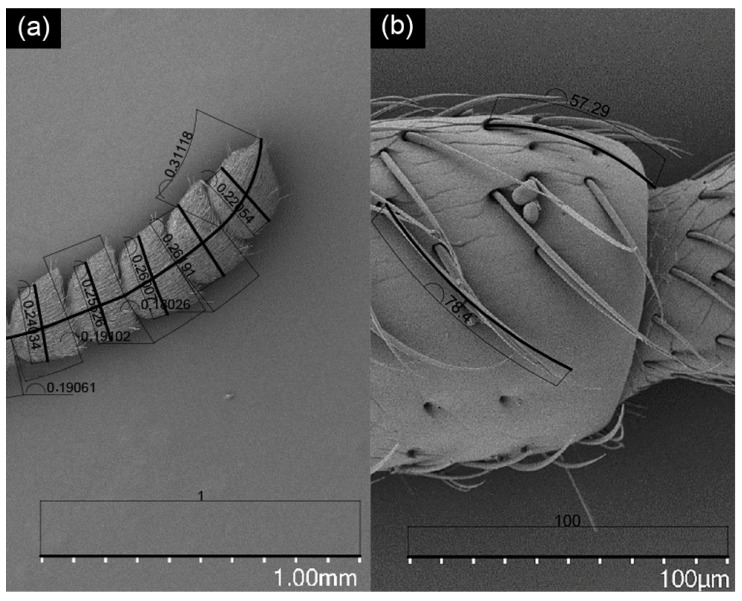
The width and length of each antenna segment (**a**) and the length of the sensilla (**b**) were measured by 2014 AutoCAD software.

## 3. Results

### 3.1. Antennae

#### 3.1.1. General Antennal Morphology

Both female and male antennae of *M. dorsalis* consisted of 11 serrated segments, including a scape, a pedicel, and flagella (F1–F9) composed of the funicle (there are seven funicular antenomers and a final distal mass) (Figure 2). The average total lengths of the male and female antennae were 2116.6 ± 67.4 μm and 2127.1 ± 57.3 μm, respectively, with the female antennae slightly longer than the male antennae. However, the antennae length and each section’s length were not significantly different between the sexes (df = 4, *t* = −0.14, *p* = 0.895) (Table 1). The widths of flagella 1 and 8 were significantly greater in females than that in males (in 1 flagellum 1, df = 4, *t* = −3.871, *p* = 0.019; in flagellum 8, df = 4, *t* = −3.248, *p* = 0.031) (Table 1).

#### 3.1.2. Types and Distributions of Antennal Sensilla

Sensilla size and shape were determined from serial SEM images, following the guidance of earlier research [13,16,22,23,24]. Twelve subtypes of sensilla on the antennae of *M. dorsalis* were observed in both sexes, including Bőhm bristles (BB), two types of sensilla trichoid (ST1, ST2), two types of sensilla chaetica (SC1, SC2), four types of sensilla basiconic (SB1, SB2, SB3, SB4), sensilla cavity (SCa), sensilla auricillica (SA), and sensilla gemmiformium (SG). Most types of sensilla were located on the flagellum. Except for the first and second flagellum, other flagella had acute-angled, wedge-shaped extensions, where great numbers of the sensilla were located. The length, average counts, and distribution of the different types of sensilla on antennae were shown in Table 1, Table 2 and Table 3. The length, average counts, distribution, and morphological features of the maxillary and labial palps are shown in Table 4.

Bőhm bristles (BB)

Bőhm bristles (BB) are triangular nail-like structures with a smooth cuticle that gradually tapers to a blunt tip, standing almost perpendicular to the antennal surface with a pitted basal socket (Figure 2b,c). They are located at the intersegmental membrane on the basal sclerite connected to the scape and between the scape and pedicel (Table 2). BB mean length in males and females was 7.5 ± 0.4 μm and 9.2 ± 0.7 μm, respectively. The average length of BB in females was significantly longer than in males (df = 58, *t* = −2.028, *p* = 0.048) (Table 3).

**Figure 2 insects-12-01112-f002:**
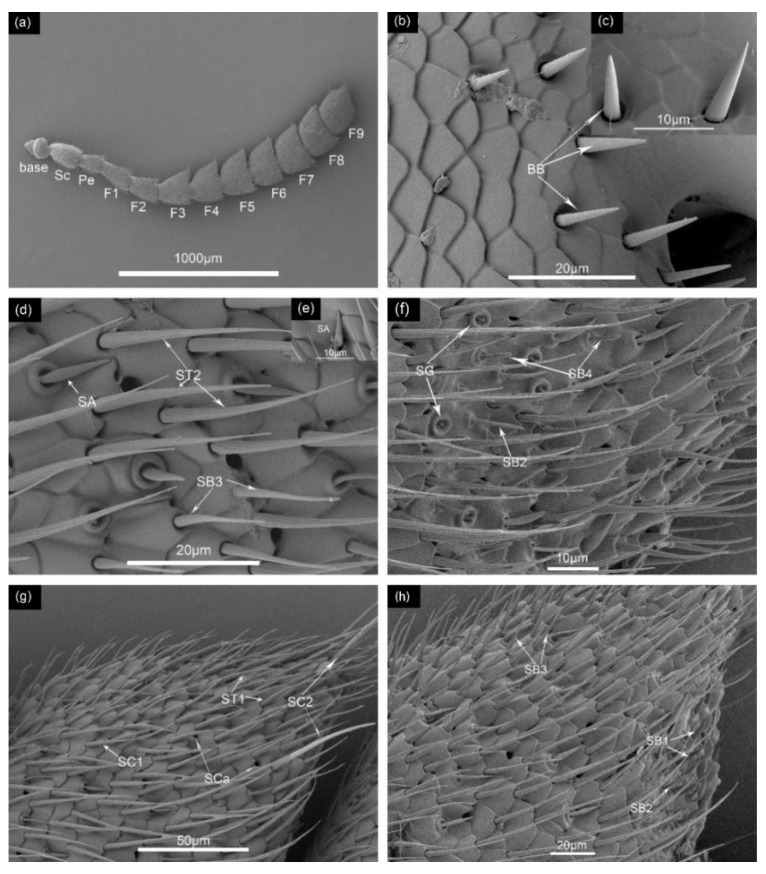
(**a**) Morphology of female *Megabruchidius dorsalis* antenna; Sc, scape; Pe, pedicel; F, flagellomereum. SEM structure of antennal sensilla of M. dorsalis. (**b**,**c**) Bőhm bristles (BB); (**d**,**e**) Sensilla trichoid (ST2), sensilla basiconic (SB3), sensilla auricillica (SA); (**f**) sensilla basiconic (SB2, SB4), sensilla gemmiformium (SG); (**g**) sensilla trichoid (ST1), sensilla chaetica (SC1, SC2), sensilla cavity (SCa); (**h**) sensilla basiconic (SB1,SB2,SB3).

**Table 1 insects-12-01112-t001:** Mean lengths of antennal segments of *Megabruchidius dorsalis* in both sexes.

Antennal Segment		Length (μm)			Width (μm)		
		Male	Female	Mean	Male	Female	Mean
Scape		198.1 ± 8.8	192.1 ± 6.5	195.1 ± 5.1	111.1 ± 2.1	130.8 ± 13.0	121.0 ± 7.3
Pedicel		155.9 ± 7.3	157.6 ± 4.3	156.7 ± 3.8	79.3 ± 1.6	89.2 ± 10.9	84.2 ± 5.4
Flagellum	F1	173.1 ± 3.9	186.3 ± 7.5	179.7 ± 4.8	78.8 ± 1.6	94.7 ± 3.9 *	86.7 ± 4.0
	F2	158.7 ± 9.8	177.7 ± 9.0	168.2 ± 7.3	89.6 ± 4.4	112.3 ± 13.8	101.0 ± 8.2
	F3	209.0 ± 5.3	199.7 ± 5.3	204.4 ± 3.9	139.3 ± 3.2	164.7 ± 10.3	152.0 ± 7.4
	F4	189.3 ± 7.5	185.9 ± 4.0	187.6 ± 3.9	186.8 ± 3.0	203.1 ± 8.9	195.0 ± 5.6
	F5	195.5 ± 11.3	186.9 ± 5.0	191.2 ± 5.8	207.5 ± 6.7	228.4 ± 6.2	217.9 ± 6.2
	F6	190.7 ± 4.9	184.7 ± 2.6	187.7 ± 2.8	222.3 ± 7.6	246.6 ± 5.8	234.4 ± 6.9
	F7	188.3 ± 8.7	184.5 ± 3.3	186.4 ± 4.2	238.7 ± 4.8	253.8 ± 4.0	246.2 ± 4.4
	F8	179.2 ± 11.2	180.2 ± 2.0	179.7 ± 5.1	235.6 ± 4.1	254.0 ± 3.9 *	244.8 ± 4.8
	F9	279.0 ± 5.2	291.4 ± 11.5	285.2 ± 6.3	203.7 ± 0.7	209.1 ± 7.3	206.4 ± 3.6
Total		2116.6 ± 67.4	2127.1 ± 57.3	2121.8 ± 33.7	-	-	-

Data are the mean ± S.E. In each row, data in the female columns, followed by an asterisk (*), indicate a significant difference between the sexes (independent-samples *t*-test; *p* = 0.05). N = 3 per sex.

Sensilla trichoid (ST)

ST1 had sharp-tipped and strong longitudinal grooves on the cuticular surfaces, which curved toward the antennal shaft, surrounded by a shallow cuticular socket (Figure 2g). ST1 sensilla were the most common and abundant type of sensilla on the antennae, distributed mostly on flagella F4 and F9. They had a mean length of 34.9 ± 0.9 μm. The females’ were slightly longer than the males’, but the difference was not significant (df = 58, *t* = −0.81, *p* = 0.421) (Table 3).

**Table 2 insects-12-01112-t002:** The distribution of sensilla on the antennae of *Megabruchidius dorsalis* adults.

Antennal Segment		BB	ST1	ST2	SC1	SC2	SB1	SB2	SB3	SB4	SCa	SA	SG
Base		39.7 ± 7.6	-	-	-	-	-	-	-	-	-	-	-
Scape		-	124.3 ± 8.1	35.3 ± 6.4	10 ± 2.8	2.2 ± 0.3	-	-	-	-	3 ± 0.8	-	-
Sc-Pe		17.3 ± 4	-	-	-	-	-	-	-	-		-	-
Pedicel		-	97.7 ± 6.9	36 ± 7.9	9.3 ± 2.3	2.7 ± 0.4	-	-	-	-	4 ± 0	-	-
Flagellum	F1	-	115 ± 10.7	34.7 ± 8.7	12.7 ± 3	3 ± 0.3	-	-	-	-	2.7 ± 0.3	-	-
	F2	-	138.3 ± 9.4	56.3 ± 13.9	16.3 ± 2.8	3 ± 0	-	-	-	-	3 ± 1	-	-
	F3	-	219.6 ± 17.2	93 ± 12	53.3 ± 3.9	4.5 ± 0.3	2 ± 1	5 ± 3	13 ± 0	-	5 ± 1.7	19 ± 5.4	-
	F4	-	260.3 ± 34.1	93 ± 8.8	88.7 ± 18.9	5.8 ± 0.3	4.8 ± 1.2	2.5 ± 0.9	12.6 ± 7.4	-	4.7 ± 0.6	30.4 ± 6.6	-
	F5	-	246 ± 37.2	111 ± 20	99 ± 8.9	5.5 ± 0.2	4 ± 0.6	4.3 ± 1.6	25.7 ± 7.3	-	6.7 ± 0.7	48 ± 9.4	-
	F6	-	216.3 ± 46.4	127 ± 16.5	151 ± 18.8	5.7 ± 0.7	7 ± 2.1	5.5 ± 2.4	68.7 ± 12.1	-	9.2 ± 0.7	58 ± 6.6	-
	F7	-	230.6 ± 45.2	128 ± 21	122 ± 11.5	6 ± 0.3	8.4 ± 1.9	5 ± 1.3	67.7 ± 12.1	-	10.2 ± 0.9	57.7 ± 10.2	-
	F8	-	238.6 ± 48.2	120.3 ± 20	140.6 ± 17.7	5.7 ± 0.3	10.5 ± 3.6	14.4 ± 6.4	76.1 ± 17.2	-	10.3 ± 1.3	60 ± 7.4	-
	F9	-	260 ± 59.3	130.3 ± 19.7	184.3 ± 20.7	7.7 ± 0.6	4.8 ± 1.4	13.7 ± 8.1	101.8 ± 21.5	12 ± 1.4	9.8 ± 0.7	67 ± 8.1	10.3 ± 0.6

Data are the mean ± S.E. Data compare means. N = 3 per sex. -, not found; Sc-Pe, the intersegmental membrane between the scape and pedicel.

In another subtype of sensilla trichoid, ST2, the bottom became thinner toward the top. ST2 possessed longitudinal ridges with a detectable depression socket, and were less curved than in ST1 (Figure 2d). For flagellum segments F1 to F9, the number of sensilla gradually ascended in both sexes of *M. dorsalis* (Figure 2). The number of ST2 were second only to the number of ST1 on the antennae and did not differ between the sexes (df = 4, *t* = 1.705, *p* = 0.163); however, the length of ST2 was 50.1 ± 1.4 μm, which is longer than the length of ST1.

**Table 3 insects-12-01112-t003:** The type, size, and shape of antennal sensilla identified in male and female *Megabruchidius dorsalis*.

Type of Sensilla	Subtype	Length (μm)					Number			
		N Sample Size	Male	Female	Mean	Median (Min–Max)	Male	Female	Mean	Median (Min–Max)
Bőhm bristles	BB	60	7.5 ± 0.4	9.2 ± 0.7 *	8.4 ± 0.4	7.8cd (2.8–16.9)	46.0 ± 9.0	68.0 ± 10.0	57.0 ± 7.8	56abcd (36–78)
Sensilla trichoid	ST1	60	34.2 ± 1.4	35.6 ± 1.4	34.9 ± 0.9	33.8ab (22.4–57.3)	2072.7 ± 533.5	2221.3 ± 198.0	2147.0 ± 256.6	2024a (1290–3092)
	ST2	60	49.1 ± 2.1	51.2 ± 1.8	50.1 ± 1.4	47.8a (32.5–78.4)	1161.3 ± 210.9	786.7 ± 61.7	974.0 ± 129.2	848ab (664–1460)
Sensilla chaetica	SC1	60	27.8 ± 1.2	29.7 ± 1.2	28.7 ± 0.8	30.2b (14.0–42.5)	1005.3 ± 93.4	769.3 ± 38.7	887.3 ± 69.5	819ab (692–1140)
	SC2	60	56.6 ± 0.9	54.5 ± 1.1	55.5 ± 0.7	55.9a (44.6–65.6)	52.3 ± 4.1	51.0 ± 1.0	51.7 ± 1.9	50.5abcd (46–60)
Sensilla basiconic	SB1	60	9.8 ± 0.3	9.9 ± 0.3	9.9 ± 0.2	9.8c (6.7–12.6)	23.3 ± 7.3	40.0 ± 8.0	31.7 ± 6.1	30.5cd (9–48)
	SB2	60	13.4 ± 0.3	12.2 ± 0.2 *	12.8 ± 0.2	12.4c (9.6–16.4)	60.0 ± 16.5	22.3 ± 6.4	41.1 ± 11.6	33.5bcd (12–90)
	SB3	60	20.3 ± 0.3	18.3 ± 0.6 *	19.3 ± 0.3	20.1bc (12.5–23.4)	367.3 ± 147.3	338.7 ± 41.0	353.0 ± 68.7	366abc (84–579)
	SB4	34	5.1 ± 0.3	6.3 ± 0.1 *	5.7 ± 0.2	5.9de (2.6–7.5)	11.3 ± 2.4	12.7 ± 1.8	12.0 ± 1.4	11cd (8–16)
Sensilla cavity	SCa	59	2.1 ± 0.1	2.2 ± 0.1 *	2.3 ± 0.1	2.1e (1.5–3.9)	65.7 ± 12.6	57.0 ± 6.6	61.3 ± 6.6	60.5abcd (44–90)
Sensilla auricillica	SA	61	11.3 ± 0.2	10.3 ± 0.3 *	10.8 ± 0.2	11.0c (1.9–13.5)	427.3 ± 13.3	230.0 ± 44.1 *	328.7 ± 48.7	361abc (180–450)
Sensilla gemmiformium	SG	20	1.3 ± 0.2	1.4 ± 0.1	1.3 ± 0.1	1.3de (0.6–2.2)	11.3 ± 0.7	9.3 ± 0.7	10.3 ± 0.6	10d (8–12)

Data are the mean ± S.E. In each row, data in the female columns followed by an asterisk (*) indicate a significant difference between the sexes (independent-samples *t*-test; *p* = 0.05). In the length columns, the lengths of sensilla trichoid, sensilla chaetica, and sensilla basiconic subtypes, marked with different lowercase letters in the same column of the same sensilla type, are significantly different (Kruskal-Wallis test; *p* = 0.05). In the number columns, the numbers of sensilla marked with different lowercase letters in the same column are significantly different (Kruskal-Wallis test; *p* = 0.05). N = 36 per sex.

Sensilla chaetica (SC)

SC1 were straight and arose from a scalelike socket to a slightly sharper tip, and the shaft surfaces had a series of longitudinal ridges (Figure 2g). For a relatively large number of SC1 sensilla, except for ST1 and ST2 (Table 3), the SC1 of flagellum segments F6 and F9 were more numerous (Table 2). The average length of SC1 was 28.7 ± 0.8 μm, with no significant difference in the sexes of *M. dorsalis* (df = 58, *t* = −1.133, *p* = 0.262) (Table 3).

Unlike SC1, SC2 had a blunt tip and straight hairs, almost perpendicular to the antennal surface axis of *M. dorsalis*. The surface of the shafts had longitudinal grooves with a hollow basal socket (Figure 2g). The number of SC2 was less than the number of ST1, ST2, and SC1 (Table 3), and SC2 was evenly distributed at the antenna segments. The length of SC2 was 55.5 ± 0.7 μm, the longest of all the sensilla types (df = 11, *p* < 0.05) (Table 3).

Sensilla basiconic (SB)

SB1 had an upright, blunt tip, with a smooth cuticle cone and a raised basal socket. The majority were situated on the lateral side of the base above the flagellum sections and at the front side of the flagellum segments (Figure 2h). They had a mean length of 9.9 ± 0.2 μm. The differences in length of SB1 were not statistically significantly between the sexes (df = 58, *t* = −0.345, *p* = 0.731) (Table 3).

The average length of SB2 was 12.8 ± 0.2 μm, which was not significantly different from that of SB1 (df = 11, *p* = 0.481). The mean length of SB2 for males and females was 13.4 ± 0.3 μm and 12.2 ± 0.2 μm, respectively, and this difference was significant (df = 58, *t* = 3.204, *p* = 0.002) (Table 3). The shape of SB2 was similar to that of SB1, i.e., a tower-shaped sensilla with a thin blunt tip (Figure 2f,h). The distribution of SB2 was analogous to SB1, being seated mainly in the lateral side of the base of the flagellum sections. SB3 was 19.3 ± 0.3 μm in length, the longest of all SB sensilla, and the male SB3 were significantly longer than those of the females (df = 58, *t* = 2.981, *p* = 0.04) (Table 3). The number of SB3 averaged 353.0 ± 68.7, the highest of all SB sensilla (Table 3).

Sensilla cavity (SCa)

The SCa was 2.3 ± 0.1 μm in the diameter; the mean diameters of the males and females were 2.1 ± 0.1 μm and 2.2 ± 0.1μm, respectively, and this was statistically significantly different (df = 58, *t* = −2.466, *p* = 0.017) (Table 3). The mean number of SCa was 61.3 ± 6.6 μm, e distributed across all segments (Table 3) (Figure 2g).

**Table 4 insects-12-01112-t004:** The type, numbers of maxillary, and labial palp sensilla identified in male and female *Megabruchidius dorsalis*.

Type of Sensilla	Location	Count			Shape	Tip	Sensillum Surface
		Male	Female	Mean			
Sensilla trichoid	Maxillary palps	38.2 ± 2.2	47.2 ± 0.7 *	42.7 ± 1.8A	Curved	Blunt	Smooth
	Labial palps	33.0 ± 4.8	34.5 ± 2.7	33.8 ± 2.6B	Curved	Blunt	Smooth
Sensilla chaetica	Maxillary palps	5.8 ± 0.3	4.7 ± 0.4 *	5.3 ± 0.3	Straight	Sharp	Smooth
	Labial palps	6.5 ± 0.4	4.8 ± 0.7	5.7 ± 0.5	Straight	Sharp	Smooth
Sensilla styloconica	Maxillary palps	20.5 ± 1.6	20.7 ± 1.5	20.6 ± 1.1A	Straight	Blunt	Top hole
	Labial palps	7.0 ± 0.4	6.2 ± 0.5	6.6 ± 0.3B	Straight	Blunt	Top hole
Sensilla coeloconica	Maxillary palps	6.0 ± 1.1	3.5 ± 0.3	4.8 ± 0.7	Straight	Sharp	Smooth
	Labial palps	5.3 ± 0.3	5.0 ± 0.6	5.2 ± 0.3	Straight	Sharp	Smooth

Data are the mean ± S.E. In each row, count values in the female and male columns followed by the asterisks (*) indicate a significant difference in number of sensilla between the sexes (independent-samples *t*-test at *p* = 0.05). N = 6 per sex. In the mean column, counts of the same sensilla type in different locations, marked with different capital letters, are significantly different (independent-samples *t*-test at *p* = 0.05). N = 6 per sex.

Sensilla auricillica (SA)

The mean SA count for males, 427.3 ± 13.3, was more than that for females (230.0 ± 44.1; df = 4, *t* = 4.281, *p* = 0.013) (Table 3). SA were located only on flagella F3 to 9F (Table 2). The SA shape resembled grass leaves, and had ear-shaped grooves and wall pores. These were thicker and shorter than the ST sensilla, obtuse at the end, and seated on a cylindrical protruding socket (Figure 2d,e). They had a mean length of 10.8 ± 0.2 μm, and the length in males (11.3 ± 0.2 μm) was significantly longer than in females (10.3 ± 0.3 μm; df = 58, *t* = 2.813, *p* = 0.007) (Table 3).

Sensilla gemmiformium (SG)

The shape of SG had a likeness to the bud of a plant seed, being short and straight, with a smooth surface and a blunt tip. SG was short, measuring 1.3 ± 0.1 μm in length (Table 3); there was no significant difference between the sexes. SG was only distributed in flagellum F9 (df = 18, *t* = −0.246, *p* = 0.809) (Table 2).

#### 3.1.3. Density of Antennal Sensilla

Among all sensilla types, ST1 was the most abundant on the antennae of both sexes of *M. dorsalis*, followed by ST2 and SC1. There were significant differences in the number of all types of sensilla in Table 3 (df = 11, *p* < 0.05). The number of SA in males was significantly higher than in females, with no significant differences between the sexes of residual types of sensilla. ST, SC, and SCa were the most widely distributed of all antenna segments; BB was located only on the base of the scape and pedicel; SB, SA, and SG were distributed on flagella F3 to F9 (Table 2).

### 3.2. The Maxillary and Labial Palps

The mouthparts of each adult *M. dorsalis* consisted of the labrum, a pair of unjointed mandibles, a pair of maxillae, the lower lip, and a hypopharynx (Figure 3a). The maxillary palps had four segments and the labial palp had three segments (Figure 3a).

**Figure 3 insects-12-01112-f003:**
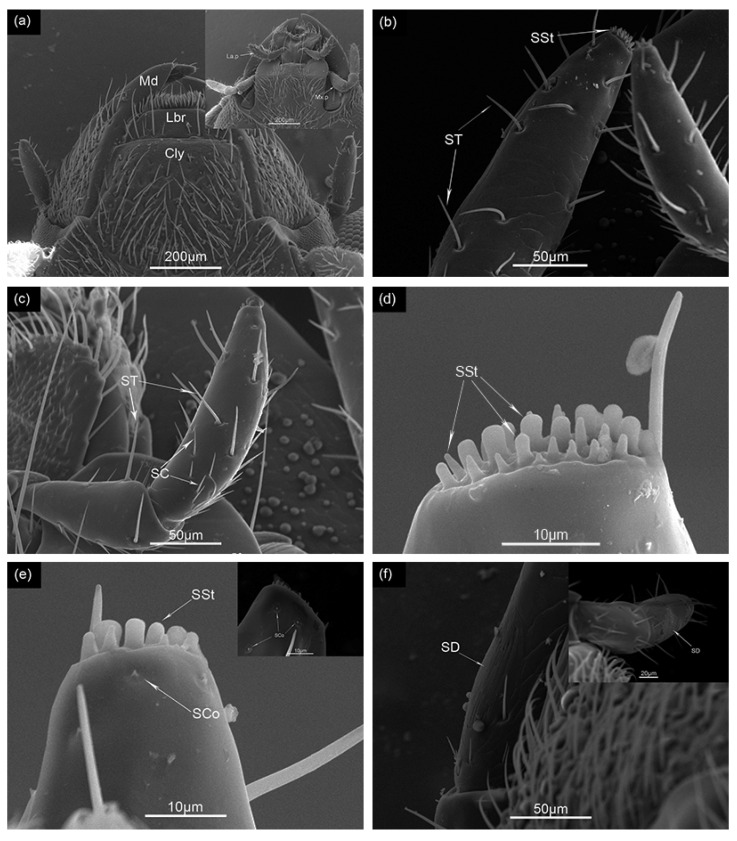
(**a**) Morphology of male *Megabruchidius dorsalis* mouthpart: Md, mandible; Lbr, labrum; Cly, clypeus; La.p, labial palp; Mx.p, maxillary palp. SEM structure of maxillary and labial palps sensilla of *M. dorsalis*: (**b**) sensilla trichoid (ST), sensilla styloconica (SSt); (**c**) sensilla trichoid (ST), sensilla chaetica (SC); (**d**) sensilla styloconica (SSt); (**e**) sensilla coeloconica (SCo), sensilla styloconica (SSt); (**f**) sensilla digitiform (SD).

#### Types and Distributions of the Maxillary and Labial Palp Sensilla on the Mouthpart

We found five types of sensilla on the maxillary and labial palps of *M. dorsalis*: sensilla chaetica (SC) (Figure 3c), sensilla trichoidea (ST) (Figure 3b,c), sensilla styloconica (SSt) (Figure 3b,d,e), sensilla coeloconica (SCo) (Figure 3e), and sensilla digitiform (SD) (Figure 3f); all sensilla types had a smooth shaft. The number and distributions of each sensillum on the maxillary and labial palps of *M. dorsalis* were shown in Table 4.

SC and ST were mainly distributed on the final palp segment and the SCo was situated only on the final palp segment. Sensilla digitiform (SD) was observed in the final maxillary palp. The amount of ST sensillum in the maxillary palps in males was significantly less than in females (df = 10, *t* = −3.83, *p* = 0.009), while the number of SC sensillum in the labial palps in males was significantly greater than in females (df = 10, *t* = 2.236, *p* = 0.049) (Table 4). The numbers of ST and SSt in the maxillary palps were significantly higher than in the labial palps (ST: df = 22, *t* = 2.819, *p* = 0.01; SSt: df = 22, *t* = 12.644, *p* < 0.05) (Table 4).

## 4. Discussion

In *M. dorsalis*, the scape, the pedicel, and the flagellomeres were similar in structure to those reported for other seed beetle species, such as *Acanthoscelides obtectus* [23]. Moreover, we identified 12 subtypes of sensilla on the antennae and five types on the maxillary and labial palps of *M. dorsalis*. No significant sexual dimorphism was observed in sensilla type or distribution. However, the sizes and numbers of different sensilla types differed greatly between males and females. The various types of sensilla on the antennae, the maxillary, and the labial palps of *M. dorsalis* were similar to those reported in other species of Bruchinae bean beetles [13,16,22,23,24].

### 4.1. Antennal Sensilla

The external morphology and distribution of BB sensilla in our study conform with those reported for many other bean weevils. In antennae, it is located at the intersegmental membrane between the scape and pedicel, and on the base of the scape, which is connected to the head [23]. BB can sense the position of the antenna and control its movement as a proprioceptor [14,27]. Here, the length of BB in male adults was greater than in female adults, which may indicate that the flight ability of males is stronger than that of females. In contrast, a sexual difference in BB abundance was not observed in *C. chinensis* and *C. maculatus* [13]. This may be related to the insect’s host plant (herb or tree), as beetles (28%), pupae (11%), and larvae of the fourth instar (61%) hibernate on harvested *Gleditsia* lying on the soil from previous years, and males with greater flight ability are more likely to mate [28]. The BB sensilla described in the present study were common sensory organs, except in Coleoptera, which coincides with previous findings of the sensilla on the antennae of the *Aphidius gifuensis* Ashmaed (Hymenoptera: Aphidiidae) [29], *Erannis ankeraria* Staudinger (Lepidoptera: Geometridae) [30], and probably in all other species as well. Furthermore, we did not notice any pores on the grooved sensilla trichoid, and previous studies have concluded that sensilla without pores are typically related to mechanoreceptor, thermoreceptor, or hygroreceptor function [31]. The external morphology of ST and SC conforms to the overall characteristics of mechanical sensilla, as a proprioceptor that perceives antennal movements and orientation in *Ips typographus* L. (Curculionidae: Scolytinae) [32]. In this study, the external morphology (varying in length, hair, and the presence of a sharp or blunt apex) and the distribution (large in number, Sc-F9) of ST on the antennae of *M. dorsalis* recorded are largely in conformity with earlier reports for other bruchids [13,14,16,22,23]. We observed that shaft surfaces of SC1 had longitudinal ridges without pores, were located on the antennae, and covered all antennae sections such that the antennae were conducive to contact with external matter as a proprioceptor, similar to the “tactile hairs” of *Bagnalliella yuccae* (Hinds) and *Frankliniella tritici* (Fitch) [14,33].

In our study, ST was the most abundant type of sensilla and had the widest distribution. ST1 was more abundant on the antennae and had sharp-tipped and strong longitudinal grooves on the epidermis, which resembled “sensilla trichodea type I,” ”sensilla trichoid 2′” and “sensilla trichodea type 2” on the antennae of *I. typographus* [32], *C. chinensis* (L.), *C. maculatus* (F.) [13], and *Callosobruchus rhodesianus* (Pic) (Coleoptera: Chrysomelidae) [16]. This might indicate that ST1 has an olfactory function. It is believed to have a chemosensory function in *Naupactus xanthographus* Germar (Coleoptera: Curculionidae) [34] and in some Hemiptera species [35], which are sensitive to the stimulus of pheromones [23]. In *M. dorsalis*, ST may be used to sense pheromones (except for sex pheromones, as there is no sex difference in the amount of ST). We observed that there were more SC in the antenna of male adults than in those of female adults. This result was contrary to the earlier result reported in *C. chinensis* and *Diaphania angustalis* Snellen (Lepidoptera: Crambidae) by Zhang et al. [36], and was found also in *C. maculatus*. This result and the slightly grooved blunt tip of SC2 could represent contact chemoreceptors to perceive and recognize host stimuli [12,23,37,38]. We found SB1, SB2, and SA usually exhibited numerous pores on their surfaces, implicating them as having an olfactory function [23,24,39]; SB3 were the longest and the most numerous of all SB sensilla; they lack an apical pore and wall pores and may be chemoreceptors that respond to odorant stimuli [24,40]. The SB4 observed in this study matched the features in other Bruchinae species (*A. obtectus*, *Callosobruchus subinnotatus, C. chinensis*, and *C. maculatus*); they have a grooved surface with a bract-like end, which indicates that this sensillum is a chemoreceptor possessing an olfactory function [16,23,24,41,42]. Nevertheless, the sexual differences in SB2, SB3, and SB4 length were not present in *C. chinensis* and *C. maculatus*, which may be due to the strict requirements of the host plant (oligophagia); in combination with their function analysis, they are likely to be able to identify the host stimulus. Until now, SA was not found in *C. maculatus*, *C.*
*subinnotatus*, or *A. obtectus*. Probably, from an evolutionary perspective, SA was either ancestral, being subequently lost in *C. maculatus*, *C. subinnotatus*, and *A. obtectus* or acquired independently in *M. dorsalis, C. chinensis*, and *C. rhodesianus* (i.e., convergence) due to a common selective force (such as similar sex pheromones). Moreover, *C. rhodesianus* is more closely related to *C. maculatus* than to *C. chinensis* based on the molecular phylogeny of the genus, and SA’s distribution resembles SCa’s in *C. rhodesianus*, *C. maculatus*, and *C. chinensis* [43]. In contrast, we believe SA is due to the same evolutionary process. SA has a shape resembling grass leaves, wall pores, and abundant sexual differences, which indicate that this sensillum has an olfactory function to detect female pheromone in this species [16,24].

The morphology of SCa, like the sensilla cavity in *C. chinensis* (L.), and their distribution, were the same as the sensilla cavitae in *C. rhodesianus* (Pic), which may indicate that this sensillum had chemo-, thermo-, or hygro-reception function [13,16]. The following causes are possible for the similarity in SCa distribution in the two bruchid species: the habitat and climate of *M. dorsalis* and *C. rhodesianus* are distributed in temperate/subtropical zones with more seasonality, so SCa may have developed an ability to sense and accommodate ambient conditions in *M. dorsalis*. SG was only distributed in flagellum 9, and resembled GP in *C. chinensis* (L.) and *C. maculatus* (F.). The likely function of these sensilla is chemo- or thermo-reception [13,16]. SG and SB4 were only distributed in flagellum F9, which imply that these were specific sensilla.

### 4.2. Maxillary and Labial Palps Sensilla

The maxillary and labial palps also have SC and ST. Similar to SC and ST located on the antennae, they likely function as chemo- and mechano-receptors [23]. The number of maxillary palps of ST sensilla in males was lower than in females, perhaps suggesting that these sensilla support host recognition and selecting the position of ovipositing (eggs were often laid on the surface of the seeds, the four walls, and top of the plastic box). The SSt in our study were similar to those of *A. obtectus* and *Melanotus villosus* (Geoffroy) (Coleoptera: Elateridae) [44], SCo in *D. angustalis* (Snellen), *C. chinensis* (L.), and *A. obtectus*. They may be thermo- and hygro-sensitive and be combined thermo- and chemoreceptors [24,36,45,46,47]. SD was also observed in the labial and maxillary palp of *C. chinensis* (L.). They are probably chemoreceptors and respond to vibratory stimulation in Coleoptera [48,49].

## 5. Conclusions

Using scanning electron microscopy, we described the morphological types and numbers of sensilla and their distributions on the antennae and mouthparts (the maxillary and labial palps) in both males and females of adult *M. dorsalis*. We hypothesized that the function of the sensilla relates to the structure of the sensilla and to insect behavior. BB can sense the position of the antenna and control its movement as a proprioceptor. ST and SC are dual-function receptors that combine mechanoreceptors and chemoreceptors. ST may be used to sense pheromones; SC can sense contact pheromones and host (*Gleditsia*) volatiles (SB also possessed the function). SA identified long-distance sex pheromones. SCa and SCo had hermo- or hygro-reception function. These hypotheses need to be further verified by single sensillum recording (SSR) and transmission electron microscopy (TEM). Our research can be used as background information for follow-up chemical ecology studies, electrophysiology, and behavioral experiments, to better understand the function of different sensors and the mechanisms associated with information on chemistry-based pest control strategies.

## Data Availability

The data presented in this study are available in the article.

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
