# Peer review of "Ultrastructure of the Sensilla on the Antennae and Mouthparts of Bean Weevils, Megabruchidius dorsalis (Coleoptera: Bruchinae)"

_insects, 2021, doi:10.3390/insects12121112_

Round 1

Reviewer 1 Report

  • Line 63: These previous studies, , what species do they refer to? Since here only some background sensilias are mentioned according to the insect under study.
  • Lines 70-74: Include the sensilias that were reported in the mentioned species to make a relation of this idea.
  • Materials and Methods: Include the diet used in the rearing of insects.
  • Line 88: ..."to the morphological characteristics"...
  • Results: -Lines 133-134: Flagellum is composed of the funicle, How many antenomers are there? Is there a final distal mass (club zone, for example)? I suggest that rather than indicating "nine flagella" (line 134), it is better to indicate how many funicular antenomers make up the flagellum.                                                                                                            -Line 161: "Bőhm bristles (BB)" appears in bold type, without bullets (I assume because of the difference with the other structures).                     -Line 169: a bullet is missing.                                                                   -Table 3: For Bőhm bristles some data appears in bold type, any reason?

I suggest a final paragraph that conclude this research and also include projections in relation TEM analysis. 

Author Response

请参阅附件。

Reviewer 2 Report

The authors present another study on antennal and mouthpars sensory equipment of a beetle (Megabruchidius dorsalis). The scanning electron microscopy images and parts of the descriptions are of surprisingly good quality.
However, all other parts of the manuscript come short for a study like this. Most importantly I don’t see a reason to publish another paper on antennal and mouthpart sensilla at all, without putting the morphology and their functions into context with what is known about other beetles. There is no functional, ecological, evolutionary or behavioural hypothesis.
Furthermore most core contents do not fulfil basic quality.
Keywords should not double with words in the title (does not contribute to indexing).
Introduction part follows important and basic papers on sensilla morphology and their probable function. However, in the last 30 years, many papers have provided more actual and precise data which are missing here, at least for an introductive presentation and summary of these structures would be welcome.
Material and methods do not sufficiently describe the methods properly. Brands and addresses are missing. The statistics are not properly described: did the data show homoscedasticity (and was this statistically tested?). Without these preassumption, the provided statistics are without value.
The SEM-figures should be improved regarding the contrast of the micrographs. Many details are lost due to strong differences in brightness and contrast.
The amount of samples investigated N=3 to N=6 is much too low for solid statistics. I wonder whether the ANOVA would be reliable at all. The test power should be reported as well. As the raw data were not included as supplementary files this can even not be validated.
The discussion does not set the results into a relevant context. Especially for the audience of Insects a broader context is required.
In conclusion, all what the authors report and describe is correct but their results are poorly put in perspective and brings almost nothing the knowledge of antenna sensilla in general and of  Megabruchidius dorsalis etho-ecology in particular. I would not recommend the paper for the journal Insects. I recommend a journal with more morphological focus or physiological focused (e.g. Physiological Entomology). 

Round 2

Reviewer 2 Report

The revised manuscript still is not suitable in Insects in my oppinion.

Firstly, the text still is full of grammatical issues (although the authors reply they consulted a service).

The remarks provided in the first round of reviews still leaves most of its validity, as in general the remarks were not really solved, but superficially tackled. it appears it is just rephrased to pretend to be amended.

The added hypothesis of the evolutionary perspective does not help, and in the way presented also does not stand: it can indeed be interesting to reflect on the role of SA in this beetle, however, the information added to the manuscript does only say, it can either be evolved in different beetles convergently, or reduced in another, without any explanation why it can eb this way, or an evaluation of the reasons for it, or even which scenario is more likely. 

I still cannot suggest to accept the manuscript. However, I see the manuscript was still given the chance to be revised. I think the isses were not solved in this revision (exept for minor formal ones, e.g. the misstype of the taxonomic affiliations and the keywords), but it's core is still very vague and not conclusive. I think the content can be, as an inventory of the sensilla equipment, in a more secialty journal (e.g. Physiological Entomology), but I wold not recommend to accept it for publication.

Author Response

Dear Reviewer,

    We would also like to express our sincere gratitude to you for your time and efforts.  Our manuscript ID: insects-1452430.

With respect to the valuable comments and suggestions themselves (in bold), the detailed modifications to our manuscript and our responses are given below in non-bold type. Our line numbers refer to the revised manuscript submitted.

Reviewer #2:Reviewer #2: Firstly, the text still is full of grammatical issues (although the authors reply they consulted a service).Thanks you for your compliments on our manuscript English language and style. Its grammar and consistency of ideas have been checked by an English language service company and we enclosed a certificate in attachment. The remarks provided in the first round of reviews still leaves most of its validity, as in general the remarks were not really solved, but superficially tackled. it appears it is just rephrased to pretend to be amended.We would also like to express our sincere gratitude to you for your time and efforts. These views have not really been addressed but have been addressed on the surface. We think it may be that we don’t understand the idea well enough. Much time was taken to carefully revise the manuscript based on the first round of reviews provided. If possible, could you make further suggestions to help us improve the article. We don’t know what the specific problem is, so we can’t make any further changes.

The added hypothesis of the evolutionary perspective does not help, and in the way presented also does not stand: it can indeed be interesting to reflect on the role of SA in this beetle, however, the information added to the manuscript does only say, it can either be evolved in different beetles convergently, or reduced in another, without any explanation why it can eb this way, or an evaluation of the reasons for it, or even which scenario is more likely. These are very clever, useful and valid points that we fully agree with. We are really grateful for freehand sketching reviewer did. The required adjustments and changes have been made. Up to now, SA was not found in C. maculatus, C. subinnotatus, A. obtectus. Probably, from an evolutionary perspective, SA was either ancestral, being lost in C. maculatus, C. subinnotatus, A. obtectus later, or acquired independently in M. dorsalis, C. chinensis and C.rhodesianus (i.e., convergence) due to a common selective force (such as the similar sex pheromone). What is more, C. rhodesianus is more closely related to C. maculatus than to C. chinensis based on the molecular phylogeny of the genus, and SA’s distribution resembles SCa’s in C. rhodesianus, C. maculatus and C. chinensis. By contrast, we prefer the difference is due to a common selective force. Please see Lines 364-367. 

I still cannot suggest to accept the manuscript. However, I see the manuscript was still given the chance to be revised. I think the isses were not solved in this revision (exept for minor formal ones, e.g. the misstype of the taxonomic affiliations and the keywords), but it's core is still very vague and not conclusive. I think the content can be, as an inventory of the sensilla equipment, in a more secialty journal (e.g. Physiological Entomology), but I wold not recommend to accept it for publication.

We are really grateful for professional and helpful comments from reviewer on introduction.

I have the following opinions on whether our paper is suitable for publication in Insects:

Frist, in this paper, we used scanning electron microscopy (SEM) to describe the morphological types and number of sensilla and their distributions on the antennae and mouthparts of both sexes of Megabruchidius dorsalis. In our study, we relate the sensor to the behavior and evolution of the bean weevil. The function of the sense of smell and taste was further clarified by means of single-sense recorder, transmission electron microscope and so on. Then, Insects has published relevant electron microscopic articles[1,2]. Finally, we would also like to express our sincere gratitude to you for your time and efforts.

Attached pleased find the following:

  1. The revised version of the manuscript ID: insects-1452430 entitled “Ultrastructure of the Sensilla on the Antennae and Mouthparts of Bean Weevils, Megabruchidius dorsalis (Coleoptera: Bruchinae)”
  2. A certificate of the English language service used for our manuscript.

Sincerely,
Dr. Wu chengxu
e-mail: [email protected]

  1. Faucheux, M.J.; Nemeth, T.; Kundrata, R. Comparative antennal morphology of Agriotes (Coleoptera: Elateridae), with special reference to the typology and possible functions of sensilla. Insects 2020, 11, doi:10.3390/insects11020137.
  2. Li, Q.; Chen, L.; Liu, M.; Wang, W.; Sabatelli, S.; Di Giulio, A.; Audisio, P. Scanning electron microscope study of antennae and mouthparts in the pollen-beetle Meligethes (Odonthogethes) chinensis (Coleoptera: Nitidulidae: Meligethinae). Insects 2021, 12, doi:10.3390/insects12070659.

Round 3

Reviewer 2 Report

The authors did not amend the issues, probably as they did not understand the feedback.

The changed part is also somewhat poor: 

"What is more, C. rhodesianus is more closely related to C. maculatus than to C. chinensis based on the molecular phylogeny of the genus" -citation of the phylogeny missing!

" a common selective force (such as the similar sex pheromone)." The sex pheromone can't be a selective pressure. This is also a result of the same evolutionary process that caused the sensillae. Selective pressures can never be anatomical/physiological features of the animal itself.

I will also repeat my initial objections, that were not amended in the first revision:

Material and methods: Brands and addresses are missing: SPSS 18.0 software

It is important to explain the investigation: the antenna has four sides, but the micrographs only show one. Were they examined from all sides, or was the facing direction of the antenna assumed to be representative for the whole circumferencial surface of the antenna?

The amount of samples investigated N=3 to N=6 is much too low for solid statistics. The study should have been based on more specimens. 3 do not reflect intraspecific variance at all. To solve this issue, at least the resulting test power should be discussed. The test power should be reported and discussed whether it is sufficient to justify the comparison. If the test power is too low, the results of the statistics are not reliable at all.

The raw data should be added as a supplementary file. 

Last of all. The authors mention a language certificate. However, there are several grammatical errors that I would not expect after a proper language editing:

e.g. l. 22 "This beetle search" is singular, should be searches, also "their" does not apply, should be "its"

l. 45f: " biological activities, including antimicrobial, antioxidant, and antischistosomals" can't be antischistosomals (antischistosomal activities)

l. 50ff: " Due to the females oviposit on the surface of seeds, and the larvae burrow into the seeds until emergence, making it challenging to implement effective chemical controls" Sentence not properly connected

l.54: "those behaviors including feeding" should be "include"

l.143: " Since the data were normally distributed homogeneity of variance" does not make sense gramatically

l. 318: "male adults are easier to fly" makes no sense gramatically

and several more

Author Response

Dear Reviewer,

    We would also like to express our sincere gratitude to you for your time and efforts.  Our manuscript ID: Insects-1452430.

With respect to the valuable comments and suggestions themselves (in bold), the detailed modifications to our manuscript and our responses are given below in non-bold type. Our line numbers refer to the revised manuscript submitted.

Reviewer #2:Reviewer #2: "What is more, C. rhodesianus is more closely related to C. maculatus than to C. chinensis based on the molecular phylogeny of the genus" -citation of the phylogeny missing!Thank you so much for your patient and thoughtful comments, the corresponding changes have been made. Please see Line 379. " a common selective force (such as the similar sex pheromone)." The sex pheromone can't be a selective pressure. This is also a result of the same evolutionary process that caused the sensillae. Selective pressures can never be anatomical/physiological features of the animal itself.This description we did is a bit confusing, and the corresponding changes have been made. Please see Lines 375-380.Referring to Fukuda et al.[1], we think that the same SA was produced by the similar sex pheromone during the evolution of these weevils. Because we infer that SA has a shape resembling grass leaves, the sexual difference in abundance, which indicates that this sensillum has an olfactory function to detect female pheromone in this species[2].

Material and methods: Brands and addresses are missing: SPSS 18.0 software I’m sorry, these are all errors caused by the author’s carelessness and have been changed. Please see Lines 145-146. 

It is important to explain the investigation: the antenna has four sides, but the micrographs only show one. Were they examined from all sides, or was the facing direction of the antenna assumed to be representative for the whole circumferencial surface of the antenna?

We are really grateful for professional and helpful comments. This is a very intelligent, valid comment that we fully agree with. Since SEM is a two-dimensional photograph, we place the antennae on a horizontal plane similar to figure 1a, and count the total number of sensors on both sides (figure 1a rotated 180°) . Mouthparts take the same approach, as shown in figure 3a. Please see Lines 132-134.

The amount of samples investigated N=3 to N=6 is much too low for solid statistics. The study should have been based on more specimens. 3 do not reflect intraspecific variance at all. To solve this issue, at least the resulting test power should be discussed. The test power should be reported and discussed whether it is sufficient to justify the comparison. If the test power is too low, the results of the statistics are not reliable at all.

We are really grateful for professional and helpful comments. The number of samples investigated N=3 to N=6, we found that Faucheux et al. [3] examined antennae of three for each sex, both antennae of three female and three male heads were observed by Vera et al. [4]. Then, This description we did is a bit confusing, we confused the initial sample (number of insects) with the number of sensor samples used for sensor data analysis, and we have changed Table 3. Please see Table 3 and description.

The raw data should be added as a supplementary file.

Thank you for your valuable advice. After careful consideration, we decided not to release the original data (which is relevant to our follow-up experiments).

e.g. l. 22 "This beetle search" is singular, should be searches, also "their" does not apply, should be "its"

Thank you for your valuable advice. The corresponding changes have been made. Please see Line 23.

l. 45f: " biological activities, including antimicrobial, antioxidant, and antischistosomals" can't be antischistosomals (antischistosomal activities)

Thank you for your valuable advice. The corresponding changes have been made. Please see Line 46

l. 50ff: " Due to the females oviposit on the surface of seeds, and the larvae burrow into the seeds until emergence, making it challenging to implement effective chemical controls" Sentence not properly connected

Thank you for your valuable advice. The corresponding changes have been made. Please see Lines 51-53.

l.54: "those behaviors including feeding" should be "include"

Thank you for your valuable advice. The corresponding changes have been made. Please see Line 55.

l.143: " Since the data were normally distributed homogeneity of variance" does not make sense gramatically

Thank you for your valuable advice. The corresponding changes have been made. All the data were normally distributed, and the variance was equal, the differences in the length and number of sensilla were analyzed by one-way ANOVA (analysis of variance), followed by a Tukey multiple comparisons posttest, and comparisons between the sexes were made using independent-samples t-tests. Please see Lines 145-154.

l. 318: "male adults are easier to fly" makes no sense gramatically

Thank you for your valuable advice. The corresponding changes have been made. Here, the length of BB in male adults was greater than in females, which may indicate that male’s flight ability is stronger than the female’s. Please see Lines 327-328.

Please note that other results that were lacking in the original submission have since been added, and a new reference are now included. Table 3 also revised.

We hope you find this version of the manuscript acceptable and look forward to hearing from you.

Attached pleased find the following:

  1. The revised version of the manuscript ID: insects-1452430 entitled “Ultrastructure of the Sensilla on the Antennae and Mouthparts of Bean Weevils, Megabruchidius dorsalis (Coleoptera: Bruchinae)”

Sincerely,
Dr. Wu chengxu
e-mail: [email protected]

References

  1. Fukuda, K.; Yanagawa, A.; Tuda, M.; Sakurai, G.; Kamitani, S.; Furuya, N. Sexual difference in antennal sensilla abundance, density and size in Callosobruchus rhodesianus (Coleoptera: Chrysomelidae: Bruchinae). Applied Entomology and Zoology 2016, 51, 641-651, doi:10.1007/s13355-016-0441-4.
  2. Wang, H.; Zheng, H.; Zhang, Y.; Zhang, X. Morphology and distribution of antennal, maxillary palp and labial palp sensilla of the adult bruchid beetles,Callosobruchus chinensis(L.) (Coleoptera: Bruchidae). Entomological Research 2018, 48, 466-479, doi:10.1111/1748-5967.12296.
  3. Faucheux, M.J.; Nemeth, T.; Hoffmannova, J.; Kundrata, R. Scanning Electron Microscopy Reveals the Antennal Micromorphology of Lamprodila (Palmar) festiva (Coleoptera: Buprestidae), an Invasive Pest of Ornamental Cupressaceae in Western Palaearctic. Biology (Basel) 2020, 9, doi:10.3390/biology9110375.
  4. Vera, W.; Bergmann, J. Distribution and ultrastructure of the antennal sensilla of the grape weevil Naupactus xanthographus (Coleoptera: Curculionidae). Microsc Res Tech 2018, 81, 590-598, doi:10.1002/jemt.23014.
